# Citation *proximus*: The role of social and semantic ties on citations

Diego Kozlowski[1]*, Carolina Pradier[1], Pierre Benz[1], Natsumi S. Shokida[1],
Jens Peter Andersen[2], Vincent Larivière[1,3,4,5]*

1 École de Bibliothéconomie et des Sciences de l'information, Université de Montréal, Montréal,
Quebec, Canada, 2 Danish Centre for Studies in Research and Research Policy, Aarhus University,
Aarhus, Denmark, 3 Consortium Érudit, Montréal, Quebec, Canada, 4 Observatoire des Sciences et
des Technologies, Centre Interuniversitaire de Recherche sur la Science et la Technologie, Université
du Québec à Montréal, Montréal, Quebec, Canada, 5 Department of Science and Innovation-National
Research Foundation Centre of Excellence in Scientometrics and Science, Technology and Innovation
Policy, Stellenbosch University, Stellenbosch, Western Cape, South Africa

* diego.kozlowski@umontreal.ca (DK); vincent.lariviere@umontreal.ca (VL)

pone.0335366

University, TAIWAN

**Peer Review History:** PLOS recognizes the
benefits of transparency in the peer review
process; therefore, we enable the publication
of all of the content of peer review and
author responses alongside final, published
articles. The editorial history of this article is
available here: https://doi.org/10.1371/journal.
pone.0335366

## Abstract

Despite being considered as key indicators of research impact, citations are shaped
by factors beyond intrinsic research quality—such as including prestige, social net-
works, and research topics. While the Matthew Effect explains how prestige accu-
mulates, our study contextualizes this by showing that other mechanisms also play a
role in citation accumulation. Analyzing a large dataset of U.S. economic (N = 43,467)
and their citation linkages (N = 264,436), we find that close ties in the collaboration
network are the strongest predictor of citations, closely followed by semantic similar-
ity between citing and cited papers. This suggests that citations are not only driven
by prestige but are strongly affected by f social networks and intellectual proximity.
Prestige remains an important factor affecting citations for highly cited papers, but
for most papers, proximity—both social and semantic—plays a more significant role.
These findings redirect focus from extreme cases of highly cited research to the over-
all citation distribution, which influences most scientists' career paths and knowledge
production. Recognizing the diverse factors influencing citations is critical for science
policy and for developing a reward system of science that is fairer and reflects a
diversity of contributions to science.

## Introduction

In the eyes of many scientists and policy-makers, citations are a key marker of
research impact. They have been used—either directly or through proxies, such as
the journal Impact Factor—for many decades to assess researchers—both in terms
of career progression [1,2] as well as funding allocation [3]—and are central compo-
nents of university rankings [4]. However, citation-based indicators have been shown

**Data availability statement:** The data underlying this study are from the Web of Science (Clarivate Analytics), a subscription-based proprietary database. The authors accessed the data through the Observatoire des sciences et des technologies (OST) database under the institution's standard subscription license. Other researchers can access the data in the same manner by obtaining access to Web of Science via an institutional or personal subscription. The data are proprietary and cannot be publicly shared; however, the authors will make available the list of article IDs from the OST with the necessary information to reproduce the analysis. The authors did not have any special access privileges.

**Funding:** This project was funded by the Social Science and Humanities Research Council of Canada Pan-Canadian Knowledge Access Initiative Grant (Grant 1007-2023-0001), and the Fonds de recherche du Québec-Société et Culture through the Programme d'appui aux Chaires UNESCO (Grant 338828). The funders had no role in study design, data collection and analysis, decision to publish, or preparation of the manuscript.

**Competing interests:** The authors have declared that no competing interests exist.

to suffer from several issues, including their lack of comparability across disciplines, their time to accumulate, as well as their skewed distributions, with a minority of papers accounting for the majority of citations [5].

Many studies have also acknowledged the multiple factors that affect papers' citations rates, which are not related with their *intrinsic* quality. Among those, the effects associated with markers of prestige—such as authors' characteristics [6–9], institutional reputation [10], or journal status [11,12]—have been heavily studied in bibliometrics, with the Matthew Effect [13]—and its mirror effect, the Matilda effect [14]—probably being the most studied [15–18]. Those markers of prestige are key drivers of papers' citations, and their cumulative nature shapes the distribution of impact, in which only a few articles and authors receive high numbers of citations, and the majority receive only a handful of citations [19]. Such power-law distributions are coherent with Bourdieu's account of prestige accumulation as a driver of symbolic capital [20], which is unevenly distributed within the scientific field.

While those previous studies highlight the *social* nature of citation links, they do not provide a complete picture of the factors that influence papers' citations. Drawing on a large database of disambiguated authors (N = 43,467) and citation linkages (N = 264,436) in the field of economics, this paper aims to contribute to this literature by analyzing how the social proximity between the citing and cited authors, as well as the semantic ties between the citing and the cited paper, relate to papers citation rates.

Social proximity refers to the degree of personal acquaintance or direct social connection between the authors of academic papers. We approximate this phenomenon through authors' co-authorship network [21], assuming that the closer two authors are in the network, the more they are likely to know each other [22–25]. In a citation context, such social ties also encompass authors' self-citations [26–30], representing the highest degree of social proximity between citing and cited authors. Including self-citations as an extreme case of social distance can be debated, since it involves a relationship with oneself. Our reasoning for including self-citations is that social proximity facilitates the literature retrieval process. We are typically more familiar with the work of colleagues we know well [31], and that includes our own. Therefore, self-citations are generally included in similar research studies [32,33], eventually leading to the presentation of collaboration networks as a broader extension of self-citation analysis [21].

Although self-citations and prestige have been well-studied, other types of social proximity—and their interplay with semantic proximity—have only been partially explored [31,34]. The role of shared past affiliations or having graduated in the same institution has also been studied in relation with co-authorship networks [35]. Previous work shows that past co-authors tend to cite their work faster [36] and more often than other authors with higher degrees of separation [32]author pairs with a distance of three or less in the co-authorship network significantly influence each other's citations [33]. This suggests that a central position in the collaboration network is associated with a larger social capital which, in turn, affects success within the citation network [22,37]. Furthermore, previous research has found a relation between proximity in the collaboration network between reviewers and authors, and outcomes in the peer review process [38].

Semantic proximity refers to the content similarity between two papers [39]. While one would expect that citing and cited papers would be thematically related [40], only recently computational methods allowed for large-scale robust comparisons of documents [41]. Some authors have also proposed the use of citations to inform the construction of semantic embeddings [42]. Despite the development of such methods, the joint effect of social and semantic proximity on citation practices remains understudied, particularly in combination with prestige.

This paper offers a novel contribution by integrating these three dimensions—social proximity, semantic similarity, and prestige—into a unified analytical framework. By combining large-scale disambiguated citation and collaboration networks with document-level semantic analysis, we provide a more holistic and empirically grounded understanding of the mechanisms behind citation practices.

We shift the focus away from highly cited outliers—which dominate much of the citation literature—and instead investigate the more frequent and ordinary citation practices that shape the careers of most scientists. This perspective is especially relevant in the context of research evaluation systems, which often rely heavily on citation metrics. Our findings reveal that social and semantic proximity—rather than prestige alone—play a central role in who gets cited. This has important implications for how scientific impact is measured and valued, particularly in policies related to hiring, funding, and promotion.

## Materials and methods

### Data

Data for this article were retrieved from the Web of Science for the period 2008–2023. Given the computational cost of working with the full bibliometric network of citations and authors, we had to restrict the analysis to a specific field and country. We aimed at building a self-contained collaboration network, and focusing on a specific field can lead to completeness problems, especially for interdisciplinary research. To mitigate these effects, we focused the analysis on the field of economics in the United States, which has a very low degree of interdisciplinarity [43], and for which social stratification has been previously studied [44,45]. To build our corpus, we use an author-based approach, and use the CWTS author disambiguation algorithm to track authors across publications [46]. A traditional journal-based approach—taking all articles published in journals from Economics—would give an incomplete list of publications for most authors (as they might publish in other related fields), which would imply an ill-defined measure of social proximity. To remedy this, we define a population of focal authors to those that contributed to at least three papers and who published at least 50% of their works within the field of Economics (as defined by the WoS) and have a US affiliation. We include all the publications of the focal authors and then expand our corpus to include all collaborators of these focal authors and their publications, a necessary step for the completeness of the collaboration network. In the appendix, we also show the results for other twelve fields that, as well as Economics, had more than 50% of their citations generated within their field, as an approximation of closeness (Earth & planetary Science, Philosophy, Orthopedics, Probability & Statistics, Meteorology & Atmospheric Science, Economics, Law, Ophthalmology, Dentistry, General Mathematics, Management, Nuclear & Particle Physics, and Astronomy & Astrophysics). Nevertheless, many of these fields are rather small and most of their authors also publish outside the field, which can create misleading results as the network is incomplete.

Our dataset includes 12,214 focal authors and all their publications, and when co-authors are considered for distance computations, it expands to 43,467 authors and 71,357 articles, of which 69,081 (97%) include abstracts.

We use the full dataset to compute social distances in the network but focus on three subsets of data for the analysis. First, we consider all pairs of documents with both an abstract and a citation link to compute semantic similarity (*citing-cited pairs, n* = 264,436). Second, we contrast this with a random sample of documents where both have abstract but with no citation link, with the condition that the potentially cited document is older than the potentially citing paper (*non-citing document pairs, n* = 257,199). Finally, for social distance, we consider all possible combinations of documents. For a set of $N$ = 71,357 nodes, there are $N(N-1)/2$ = 2,545,875,046 potential pairs, however, only 35% ($n$ = 882,777,301) of these pairs can

be observed belonging to the same network component and therefore have a non-infinite distance. We restrict the rest of the analysis—namely the semantic similarity and the model—to the sample of citing-cited pairs and the negative sample of non-citing document pairs, given the computational complexity, and to create a meaningful sample for the model.

## Methods

Our goal is to understand how social proximity, semantic similarity, and prestige relate to citing practices. We use document pairs as our unit of analysis and define and operationalize these three concepts as follows.

Social proximity is defined as the degrees of separation between authors. We operationalize this concept using the collaboration network, as authors develop knowledge of and trust in their colleagues' work through collaboration. Direct collaborator pairs are closer in social proximity than authors at a higher degree of separation. This operationalization is limited as there are other ways of social proximity, such as working at the same institution, networking interactions on conferences or social events, or ties outside academia, that are not considered. Self-citations, within this context, are an extreme case in terms of proximity, and can be naturally represented in the collaboration network as a distance of zero. Degrees of separation are defined for any pair of authors. Since our unit of analysis is pairs of documents, we define the distance between two documents as the minimum distance between any pair of authors of each of the documents, or the distance between two documents $A$ and $B$ is $D(A,B) = \min(d(a_i, b_j))$, where $d(a_i, b_j)$ is the distance between author $i$ of document $A$ and author $j$ of document $B$. Similar approaches that measure the social distance between two papers as the minimum distance between the groups of their authors have been used in previous literature [21,32]. Alternatively, we have also tested the use of the average distance between authors of the papers' dyad and obtained consistent results.

Semantic similarity is used as a proxy for content similarity between two documents. We compile articles' titles and abstracts to create their embedding representation. This means, the article is projected as a point in a geometrical space. Unlike simple keyword matching, embeddings capture context and meaning, so if two articles use different words that refer to similar meanings—such as *investment*, *capital allocation*, or *asset management*—they will be relatively close in the embedding space. We use the state of the art "multilingual-e5-large-instruct" pre-trained language model [47] as a zero-shot classifier to build an embedding of 1024 dimensions. This model was chosen instead of more traditional models such as BERT [48] or models tailored for scientific documents such as sciBERT [49] because at the moment of training it was the best performing model at Huggingface leaderboard (huggingface.co/open-llm-leaderboard/). Other possible models like SPECTRE [42] are based on the citation network, which would create an information leak with our dependent variable. We then apply the cosine similarity between pairs of articles to find how close they are in this embedding space. Cosine similarity calculates the angle between two vectors rather than their raw distance, which is useful because points in the embedding have different magnitudes, but their relative orientation in space matters more for similarity.

We define prestige as the total number of citations accumulated throughout authors' entire career. As in the case of social proximity, since our unit of analysis is document pairs, we consider the author with the highest number of citations as a prestige symbol for any given document [45]. The argument here is that the highest cited author is in general the most likely to attract prestige-based citations. In this case, nevertheless, the relation between documents is not symmetrical, so we define prestige based on the authors in the cited article, as our research question focuses on the role of prestige in the decision to cite an article.

To compare the differential effect of these three factors, we build a logistic generalized linear mixed model (GLMM) that predicts the probability of a pair of documents having a citation link given their social and semantic proximity, and the prestige of the authors of the potentially cited document. The degrees of separation in the collaboration network are considered as a series of dummy variables from distance 0 –self-citation– to "6 or more" (see appendix S5 Fig). This rescaling of distance allows us to equalize unconnected documents to the furthest distance in the network, instead of discarding these cases. Interaction terms are included as we expect that for further distances, content similarity and prestige will become more relevant. Finally, as we are not considering all factors that affect citations, such as quality, gender [50], or

institutional belonging [51], we include random intercept at the cited article level to control for these and other potential factors. We also standardize the independent variables to simplify interpretation. The model is defined as follows:

$$\text{logit}\left(P(y=1)\right) = \beta_0 + \beta_1\text{Similarity} + \sum_{i=0}^{5}\beta_{i+2}\text{Step}_i + \beta_8\text{Prestige}$$

$$+ \sum_{i=0}^{5}\beta_{i+9}(\text{Similarity} \times \text{Step}_i) + \beta_{15}(\text{Similarity} \times \text{Prestige})$$

$$+ \sum_{i=0}^{5}\beta_{i+16}(\text{Step}_i \times \text{Prestige}) + (1|article) + \epsilon$$

Where Similarity is the semantic similarity, $Step_i$ is a dummy variable for distance $i$, with the reference category being 6 or more degrees of separation. Prestige is the maximum number of accumulated citations by an author of the cited paper, excluding that paper.

We use the Average Marginal Effects (AME) of each of the independent variables to understand their relation with citations, which can be understood as the expected variation in the probability of a citation for a one-unit change in the covariate, holding all other variables constant at their mean. To illustrate the interactions between independent variables, we created a grid of values representing different combinations of covariate pairs, while keeping all other variables fixed at their means. We then used this grid to predict the model's outcome, providing a clear visualization of how the covariates interact.

## Results

### Social and semantic distances

When comparing pairs of citing and cited documents with all connected pairs of articles (i.e., pairs of documents which have a non-infinite distance in the collaboration network), notable differences emerge in terms of their social and semantic distances. Fig 1A illustrates the distribution of articles based on their degree of separation within the collaboration network. The general distribution (blue) follows a Gaussian curve centered around six degrees of separation between the authors of two articles. In Economics, this is the most typical distance, which aligns with Milgram's small-world theory [52]. At smaller distances, smaller ego-networks involve fewer authors and, consequently, fewer publications. For example, a distance of 0 indicates that both publications share at least one author, so the share of publications of distance zero is bounded by the previous publications of the authors, which is a small proportion of all the publications in the field.

The distribution of citing-cited pairs (red), however, shows a different pattern. The proportion of self-citations varies significantly between these two sets. While a distance of 0 is negligible among random article pairs, it accounts for nearly 20% of citation pairs. For citing-cited pairs, the distribution is bi-modal: beyond self-citations, the distribution is centered around four degrees of separation, rather than six. Overall, the distribution of citing-cited pairs is shifted towards closer distances and displays a left-skewed shape.

Fig 1B presents the distribution of semantic similarity between articles. Due to computational complexity, to analyze non-citing document pairs we used a sample size equivalent to that of the citing-cited pairs. While the absolute values of cosine similarity are not directly interpretable—as they depend on the embedding space used to project the documents—, comparisons between groups remain meaningful [53]. The results reveal that non-citing document pairs exhibit a more symmetrical and leptokurtic distribution, centered around lower similarity values. In contrast, citing documents pairs show higher semantic similarity.

When analyzing the combined distribution in Fig 1C, we see that the random sample of non-citing document pairs shows a more leptokurtic distribution. While self-citations tend to have greater semantic similarity and exhibit higher dispersion, there is no strong correlation between social distances and semantic similarity when examining each group separately. Citing-cited pairs are closer both socially and semantically; however, these two patterns operate independently of each other.

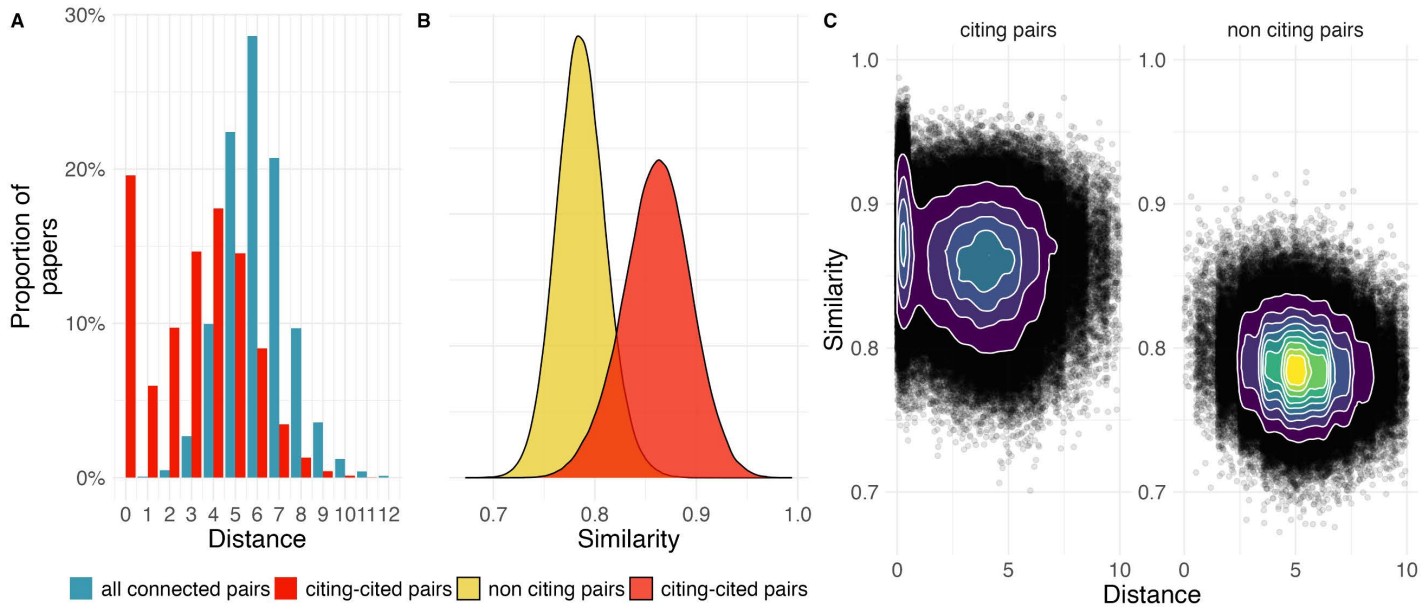

**Fig 1. Distribution shift of citing papers.** (A) Distribution of degrees of separation in the collaboration network for the citation pairs and all paper pairs with a non-infinite distance. (B) Distribution of cosine similarity of the embedding representation of citing-cited pairs and a random sample of non-citing document pairs. (C) Combined distribution of distance and similarity for citing-cited pairs and non-citing document pairs.

Observing the distribution of distances by deciles of authors' citations (see S1 Fig) reveals that prestige significantly influences the distribution. Less cited authors tend to have a larger proportion of self-citations, while highly cited authors gather more citations from documents separated by three to four degrees. This finding is aligned with the Matthew Effect, as the articles of less prestigious authors do not have a cumulative citation advantage from their peers, so they are mostly cited by themselves. Moreover, there is a lower bound in the number of self-citations that an author can receive based on the total number of papers they published, and highly cited authors naturally go beyond this bound. Additionally, due to their more extensive collaboration network, more prestigious authors tend to be more central in the network, with their papers being most frequently at five rather than six degrees of separation of all other papers. This aligns with previous studies that show how centrality in the collaboration network is associated with prestige and accumulated citations [25]. On the other extreme, the least prestigious authors have a larger than average distance to other authors.

When considering other fields (see S2 Fig) a commonality in the main patterns described above can be observed, with a high relevance of self-citation and left skewness among citing-cited pairs. Nevertheless, we can also observe that some smaller fields show very small networks, which might be an indication of the problems of building a self-contained network of co-authorship around small disciplines. On the other hand, semantic similarity (see S3 Fig) does not carry this construction problem and shows very similar patterns across disciplines, and similar degrees of overlap between the distributions of citing and non-citing document pairs.

## The relation between proximity and citations

The varying distances observed between cited pairs and others raise the question of how these three elements —social distance, semantic proximity, and prestige— influence the decision of citing an article. Fig 2 shows the results of a GLMM logistic regression that predicts the probability of a citation between a pair of articles, considering the influence of these three elements, their interactions, and a random effect at the cited article level to control for remaining unseen factors, such as the intrinsic quality of each cited article (see Data and Methods).

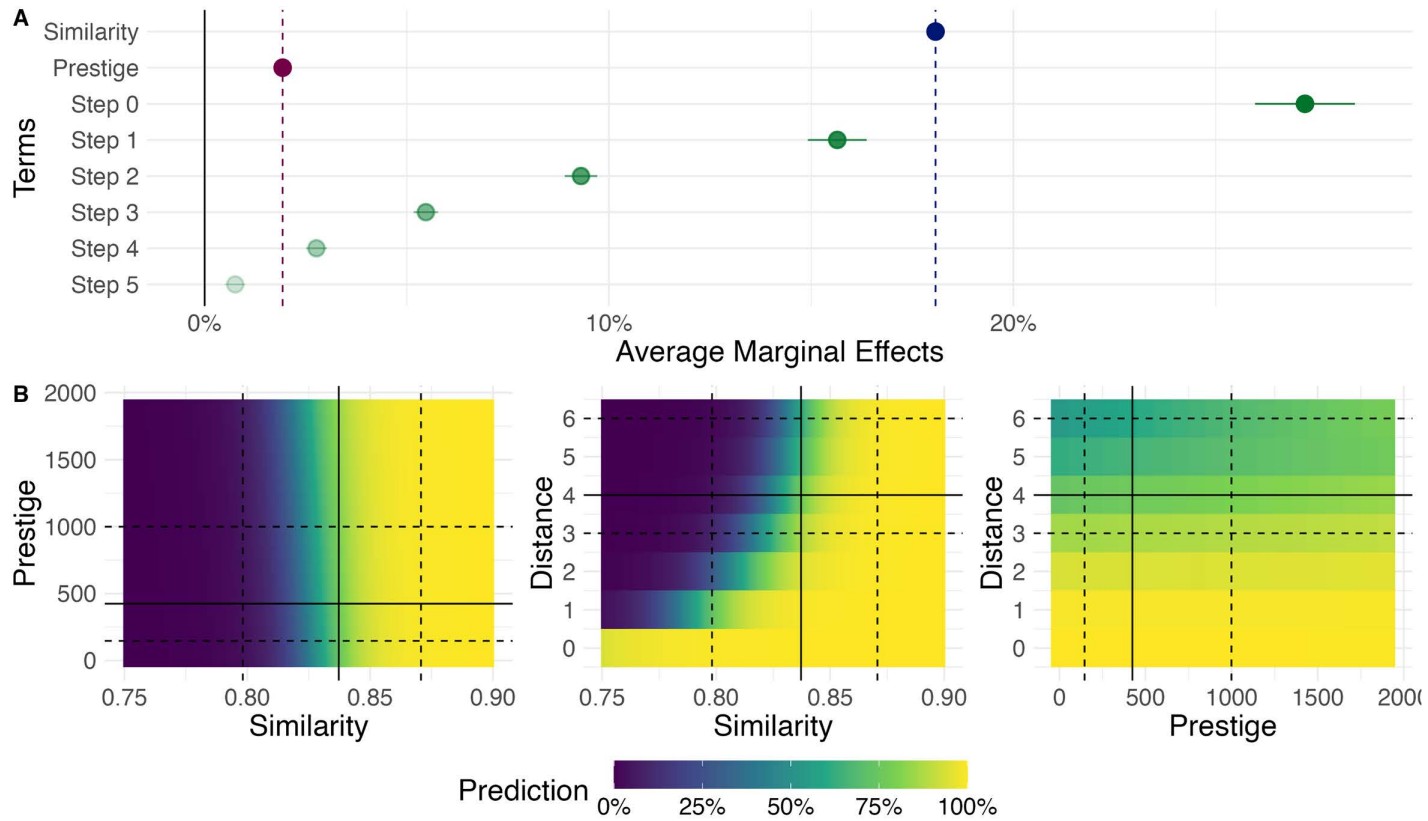

**Fig 2. The effects of social distance, semantic similarity, and prestige on citation links.** A) Average Marginal Effects of cosine similarity between papers, accumulated citations of cited authors and distances on the collaboration network on the existence of a citation link. B) Predicted probabilities at the interaction of independent variables. Solid lines represent the median of each variable, while dashed lines represent the first and third quartile of their distribution.

The results show striking differences between these three explaining factors. Fig 2A shows the average marginal effect (AME) of each of the independent variables. There, we can see that shared authorship–self-citation–has the largest average marginal effect. If the two documents share an author, there is a 25% greater chance that a citation occurs with respect to a document authored by people at a distance of 6 or more degrees of separation. If the paper was written by a direct co-author, the chances of a citation increase by more than 10%, and this effect decreases as we move further away in the collaboration network. Semantic similarity between documents has a very large AME, at the middle point between self-citations and direct collaborations and relates directly to the role of citations. Continuous variables were standardized prior to model fitting, meaning that each variable was rescaled to have a mean of 0 and a standard deviation of 1. As a result, the AME associated with semantic similarity represents the expected change in the outcome when semantic similarity increases by one standard deviation. In this context, the effect size of semantic similarity lies between those of self-citations and direct collaborations, suggesting that a one-standard-deviation increase in similarity has an impact on the outcome that is intermediate in magnitude compared to these two factors. The fact that citations are more often between semantically related works, after controlling for other social factors, is also a validation of the epistemic function of citations. This is, citing a document is not just a social phenomenon. In this respect, the strength of prestige as a predictor is revealing in the sense that it contextualizes the Matthew effect. Although significantly above 0, the effect of the total citations of the cited authors is marginal with respect to the effect of social and semantic proximity, being smaller than

the AME of 4 degrees of separation. This means that in our model, being the co-author of the co-author of the co-author of the co-author of a paper seems to have a higher influence than being a very highly cited author in the field. It is worth noting that this result is relevant to understanding unique citation links. However, cumulative effects must be considered to understand aggregate results. While the prestige of an author is defined equally across all articles in the field, social and semantic distances are unique to each pair of documents, and therefore prestige will carry greater cumulative effects than these other elements. When we run an equivalent model on other fields (see S4 Fig) we observe both regularities and variations. Across all fields, a zero-distance in the collaboration network (self-citations) is a very important predictor of a citation link. In some fields such as Dentistry, Ophthalmology, Orthopedics, or Meteorology, further steps in the collaboration network are not as important, as we observe a discrete jump in the AME of step 0 with respect to steps 1–5. In other fields such as Mathematics, Probability, Management or Education, there is a smooth decrease in the AME of the different steps, forming a ladder of decreasing importance. The interpretation of these differences should be done with caution, as the discrete jumps observed in some fields could also be an indication of ill-defined collaboration networks, as size differences can be observed on the distribution of distances in the networks (see S2 Fig). Semantic similarity consistently plays a very significant role and ranks among the top AME of each model. Astronomy and Astrophysics in particular are an exception, as they have an ill-defined model for collaboration distances. This can likely be explained by the minimal overlap in the distributions of citing and non-citing document pairs within the collaboration network (see S2 Fig). As an alternative operationalization, S6 Fig shows the results for a model with the average distance (rounded and discretized to keep a similar model structure) between the authors of the two documents instead of the minimum.

The results remain robust despite this variation, although some expected differences emerge. Since step 0 corresponds to a self-citation for all authors, the proximity between the two documents is even greater, which also results in a higher AME value. In general, because in this alternative model each step indicates a closer proximity, all the AME values tend to be higher.

The interactions between these three different covariables of citations are also revealing. Fig 2B shows the predicted probability of a citation between two articles for combinations of different values of social distance, semantic similarity, and prestige. If we focus on the interaction between prestige and similarity (left), we can observe that prestige plays a marginal role, only at similarity levels between the first quartile and the median. If similarity is significantly below or above the median, the probability of a citation has no relation to prestige. On the other hand, the relation between semantic similarity and social distance (middle) shows a stronger interaction, except in the case of self-citations. For self-citations, the probability of a citation always remains high despite variations in semantic similarity. For all other cases, we observe a diagonal that defines the shift in the probability of citation. Finally, in the relation between social distance and prestige (right), we can see both an absolute gradient for distance, but also a small effect of prestige for degrees of separation four and higher. Overall, we observe that prestige only seems to be relevant for more distant papers, both semantically and socially.

## Discussion

In this study we analyze the relationship between citations and social proximity, semantic similarity, and prestige. Our results show that the biggest predictor of a citation is shared authorship. Although we do not examine the causality behind self-citations, this phenomenon could be interpreted in different ways. It may reflect non-ethical practices, or it could simply be due to authors being more familiar with their own work than anyone else's. Additionally, authors tend to work on similar topics and research lines, making their previous work more accessible and appropriate for citation [29]. The second most important predictor of citations is the semantic similarity between documents. This means that citations do have a meaning beyond social relations. We do not only cite because of a social construct, but because previous work is related to ours. Within the social factors that shape citations' decisions, social proximity is many times more important than the prestige of cited authors. While the cumulative properties of prestige make it a relevant factor to understand the shape of the distribution of citations, there are other social aspects that are more important to understand why a paper

gets cited. These results give a new dimension for the analysis of inequalities in academia. Non-meritocratic factors are not only based on the Matthew effect. Working on mainstream topics will imply a higher similarity with several potential citing articles. Having a strong network of collaborators will also increase the chances of being cited. Only after considering these two elements is that the Matthew effect becomes relevant to explain most of the citations.

In other terms, while prestige is key for understanding highly cited papers, our results indicate that the reasons behind most citations of non-highly cited papers might be beyond prestige, and closer to semantic and social proximity, among other factors. Shifting the focus from highly cited articles towards the bulk of the distribution of citations creates a space of inquiry that can explain the phenomena that affect most scientists. Differences in impact at the middle of the distribution can have a heavy influence in career development for most scientists, which can in turn affect what science is produced.

These results are limited to economics in the US, our analysis on other fields showed comparable results on semantic similarity (S3 Fig) with all fields, and more variability on social proximity (S2 and S4 Figs), given the complexity of building a comprehensive collaboration network on a single field. This project opens several avenues for future research. Combining collaboration and citation networks with textual analysis offers opportunities to uncover new insights and raise further questions within the field of the science of science. One major challenge is to build a global collaboration network that includes all authors and articles across disciplines and countries to accurately compute distances between authors. Adding other factors such as institutional affiliation and journal would also give a more nuanced picture of prestige, which would provide a deeper understanding and enhance the findings presented in this study.

## Supporting information

**S1 Fig. Distribution of distance between articles by deciles of authors' citations on citing-cited pairs and all pairs of articles.**
(TIF)

**S2 Fig. Distribution of distances over different fields.**
(TIF)

**S3 Fig. Distribution of semantic similarity over different fields.**
(TIF)

**S4 Fig. Average Marginal effects on different fields.**
(TIF)

**S5 Fig. Performance metrics of models including different cut-points of degrees of separation.** Each cut-point represents a model with that number of dummy variables, where the last category also includes all further distances and is the reference value. The horizontal line represents the model with distance as a continuous variable. Models including only a self-citation flag (cut-point 1), and co-authors flag (cut-point 2) underperform with respect to the continuous model, but models including up to 4–6 degrees of separation show an improvement with respect to the continuous version. After 6, the improvement of the models are marginal.
(TIF)

**S6 Fig. The effects of social distance, semantic similarity, and prestige on citation links using average distance between co-authors of paper dyads.** A) Average Marginal Effects of cosine similarity between papers, accumulated citations of cited authors and distances on the collaboration network on the existence of a citation link. B) Predicted probabilities at the interaction of independent variables. Solid lines represent the median of each variable, while dashed lines represent the first and third quartile of their distribution.
(TIF)

## Acknowledgments

The authors wish to thank Maxime Holmberg Sainte-Marie for his participation in an earlier phase of the project

## Author contributions

**Conceptualization:** Diego Kozlowski, Carolina Pradier, Pierre Benz, Natsumi S. Shokida, Jens Peter Andersen, Vincent Larivière.

**Data curation:** Diego Kozlowski.

**Formal analysis:** Diego Kozlowski.

**Funding acquisition:** Vincent Larivière.

**Investigation:** Diego Kozlowski, Carolina Pradier.

**Methodology:** Diego Kozlowski, Carolina Pradier, Pierre Benz, Natsumi S. Shokida, Jens Peter Andersen, Vincent Larivière.

**Project administration:** Vincent Larivière.

**Resources:** Vincent Larivière.

**Software:** Diego Kozlowski.

**Supervision:** Jens Peter Andersen, Vincent Larivière.

**Validation:** Diego Kozlowski, Carolina Pradier, Pierre Benz.

**Visualization:** Diego Kozlowski.

**Writing – original draft:** Diego Kozlowski, Jens Peter Andersen, Vincent Larivière.

**Writing – review & editing:** Diego Kozlowski, Carolina Pradier, Pierre Benz, Natsumi S. Shokida, Jens Peter Andersen, Vincent Larivière.

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
