## [Decision Letter · Decision Letter 0]

15 Aug 2025

Dear Dr. Kozlowski,

Thank you for submitting your manuscript to PLOS ONE. After careful consideration, we feel that it has merit but does not fully meet PLOS ONE’s publication criteria as it currently stands. Therefore, we invite you to submit a revised version of the manuscript that addresses the points raised during the review process.

We look forward to receiving your revised manuscript.

Kind regards,

Mu-Hsuan Huang

Academic Editor

PLOS ONE

Journal Requirements:

2. Thank you for stating the following financial disclosure: [This project was funded by the Social Science and Humanities Research Council of Canada Pan-Canadian Knowledge Access Initiative Grant (Grant 1007-2023-0001), and the Fonds de recherche du Québec-Société et Culture through the Programme d'appui aux Chaires UNESCO (Grant 338828)]. 

3. Thank you for stating the following in the Acknowledgments Section of your manuscript: [This project was funded by the Social Science and Humanities Research Council of Canada Pan-Canadian Knowledge Access Initiative Grant (Grant 1007-2023-0001), and the Fonds de recherche du Québec-Société et Culture through the Programme d'appui aux Chaires UNESCO (Grant 338828)]. 

Please remove any funding-related text from the manuscript and let us know how you would like to update your Funding Statement. Currently, your Funding Statement reads as follows: [xxx]

4. For studies involving third-party data, we encourage authors to share any data specific to their analyses that they can legally distribute. PLOS recognizes, however, that authors may be using third-party data they do not have the rights to share. When third-party data cannot be publicly shared, authors must provide all information necessary for interested researchers to apply to gain access to the data. (https://journals.plos.org/plosone/s/data-availability#loc-acceptable-data-access-restrictions)

5. We notice that your supplementary figures are uploaded with the file type 'Figure'. Please amend the file type to 'Supporting Information'. Please ensure that each Supporting Information file has a legend listed in the manuscript after the references list.

6. We notice that your supplementary figures are included in the manuscript file. Please remove them and upload them with the file type 'Supporting Information'. Please ensure that each Supporting Information file has a legend listed in the manuscript after the references list.

Reviewers' comments:

Reviewer's Responses to Questions

**Comments to the Author**

1. Is the manuscript technically sound, and do the data support the conclusions?

Reviewer #1: Partly

Reviewer #2: Partly

2. Has the statistical analysis been performed appropriately and rigorously?

Reviewer #1: Yes

Reviewer #2: Yes

3. Have the authors made all data underlying the findings in their manuscript fully available?

Reviewer #1: No

Reviewer #2: Yes

4. Is the manuscript presented in an intelligible fashion and written in standard English?

Reviewer #1: Yes

Reviewer #2: Yes

Reviewer #1: The manuscript presents a study of the influence of three distinct factors - social ones as well as cognitive ones - on the likelihood of citations. The studied factors are social proximity of authors, semantic similarity of paper content, and author prestige. Such a study is both interesting and important due to the role of citations and citation indicators in research evaluation in general. This motivation is well argued by the authors.

A further strength of the research is the analytic framework, multivariate analysis via generalized linear model, which is appropriate to the data and the research question.

However, in my opinion, the current manuscript version has some weaknesses that ought to be addressed in a revision.

There is no overview of the current state of knowledge on the influence of social and cognitive factors on citations. Such a literature review is very much needed for readers to be able to assess the contribution of the present paper.

Moving on to the Methods and Data section, the authors mainly study a dataset of publications by economists. In the abstract this is qualified as US economists but in the main text this restriction to the United States is not mentioned except in the last paragraph of the paper. This should be clarified. The supplemental material extends the analysis to selected disciplines. Regarding these, the final section states "These results are limited to economics in the US and differences by field can be observed". Here it is unclear if this is supposed to mean that none of the results could be reproduced in the other disciplines or what the variations are.

The authors have chosen one particular operationalization of social proximity, the distance between authors in the co-authorship network. They mention other possible operationalizations ("there are other ways of social proximity, such as working at the same institution, networking interactions on conferences or social events, or ties outside academia, that are not considered"). Their chosen operationalization certainly makes sense in terms of face validity and in feasibilty of data collection. However, it is very important to proffer convincing evidence that this indicator also covers the construct of social proximity sufficiently well to be useful for the study. This could be done by discussing the literature on academic social proximity, its ideal measurement operationalization, and the comparative performance of the co-authorship network measure, if such literature exists. The authors could additionally demonstrate the robustness of their method by showing that a minor variation in implementation does not alter the result patterns. The present implementation considers only the closest author pair connection but there are obvious alternatives such as the average or the sum of all observed pairs.

Also, an argument could be made against the inclusion of self-citations into the variable for social proximity as there is nothing social about one's identity with oneself. That is to say, it is not logical that self-citations are one extreme of a continuous spectrum from close to distance social relationships.

For the prestige measure, the authors use the total citation count of the most cited author of a (cited) paper. In my opinion, there might an issue of dependence between this predictor and the independent variable, the presence/absence of a citation link, because both variables are calculated from the same citation network. For a more generally highly cited author, it would appear that also any given paper would have higher chance of being cited by the given citing paper in a sampled paper pair. This would bias the analysis. I would like to ask the authors to address this possible concern.

It might be interesting to compare and constrast the obtained results with findings of research that studied the influence of similar constructs on the outcomes of peer reviews.

Reviewer #2: This study investigates how factors beyond intrinsic research quality—such as prestige, social proximity, and semantic similarity—influence citation patterns. Using a large dataset of disambiguated authors and citation links in U.S. economics, it finds that collaboration ties are the strongest predictor of citations, followed by semantic similarity. While prestige explains highly cited papers, most citations are driven by intellectual and social proximity. These findings highlight that citation inequalities stem more from network structures and research topics than from cumulative advantage, offering critical insights for science policy and evaluation.

This study integrates social proximity, semantic similarity, and prestige into a unified analytical framework. By combining large-scale disambiguated citation and collaboration networks with document-level semantic analysis, it offers a more comprehensive and empirically grounded understanding of citation mechanisms. The findings have important implications for the measurement and evaluation of scientific impact, particularly in policies related to hiring, research funding allocation, and academic promotion.

The paper could be further improved in the following aspects:

1. The paper needs to provide a precise definition of its core concepts. The term citing behaviour encompasses not only citation outcomes but also various aspects such as citation motivations. Therefore, it is necessary for the paper to clearly delineate the scope of this concept.

2. To manage the computational cost of processing the full citation and author networks, the analysis is limited to the field of economics. However, this raises concerns about the representativeness of the findings—are the conclusions drawn from this dataset also applicable to other disciplines with low interdisciplinarity? Moreover, in highly interdisciplinary fields, do the relationships between citations and factors such as social proximity, semantic similarity, and prestige differ significantly from those in less interdisciplinary domains? The paper could benefit from including empirical analyses across more fields.

3. The study uses multilingual-e5-large-instruct as a zero-shot classifier to generate 1024-dimensional embeddings for document representation. It remains unclear why other popular text embedding methods, such as BERT or SciBERT, were not considered. A comparison or justification for this choice would strengthen the methodological rigor.

4. The paper defines prestige as the total number of citations accumulated throughout an author’s career, using the most-cited author of the cited paper as the representative of prestige. However, prestige is also commonly associated with factors such as institutional affiliation or the receipt of prestigious awards. Therefore, the definition of prestige could be expanded to include additional dimensions for a more nuanced analysis.

5. In the abstract, the study refers to "a large dataset of disambiguated authors (N=43,467) and citation linkages (N=264,436) in U.S. economics," but the main text does not clearly specify that the dataset is confined to the U.S. economics domain. A more explicit clarification of this in the main body of the paper would improve transparency and consistency.

**Do you want your identity to be public for this peer review?** For information about this choice, including consent withdrawal, please see our Privacy Policy

Reviewer #1: No

Reviewer #2: No

---

## [Author Response · Author response to Decision Letter 1]

23 Sep 2025

Reviewer #1: The manuscript presents a study of the influence of three distinct factors - social ones as well as cognitive ones - on the likelihood of citations. The studied factors are social proximity of authors, semantic similarity of paper content, and author prestige. Such a study is both interesting and important due to the role of citations and citation indicators in research evaluation in general. This motivation is well argued by the authors.

A further strength of the research is the analytic framework, multivariate analysis via generalized linear model, which is appropriate to the data and the research question.

However, in my opinion, the current manuscript version has some weaknesses that ought to be addressed in a revision.

Response: We would like to thank the reviewer for positive comments and the detailed suggestions. We believe that the changes we made based on these comments have strengthened the manuscript.

There is no overview of the current state of knowledge on the influence of social and cognitive factors on citations. Such a literature review is very much needed for readers to be able to assess the contribution of the present paper.

Response: We agree that a more detailed literature review was needed. After a careful search, we find that most of the work studying the influence of social factors such as the collaboration network remains at the aggregate level properties (i.e. centrality in the network), and not at the specific distance between the authors of a dyad of documents. Nevertheless, we extended the introduction with the following:

“Although self-citations and prestige have been well-studied, other types of social proximity—and their interplay with semantic proximity—have only been partially explored (Abramo et al., 2020; Milard & Tanguy, 2018). The role of shared past affiliations or having graduated in the same institution has also been studied in relation with co-authorship networks (D’Ippoliti et al., 2023). Previous work shows that past co-authors tend to cite their work faster (Zingg et al., 2020) and more often than other authors with higher degrees of separation (Martin et al., 2013)author pairs with a distance of three or less in the co-authorship network significantly influence each other’s citations (Singh et al., 2020). This suggests that a central position in the collaboration network is associated with a larger social capital which, in turn, affects success within the citation network (Abbasi et al., 2014; Sarigöl et al., 2014). Furthermore, previous research has found a relation between proximity in the collaboration network between reviewers and authors, and outcomes in the peer review process (Dondio et al., 2019)”

…. “Some authors have also proposed the use of citations to inform the construction of semantic embeddings (Cohan et al., 2020).”

Moving on to the Methods and Data section, the authors mainly study a dataset of publications by economists. In the abstract this is qualified as US economists but in the main text this restriction to the United States is not mentioned except in the last paragraph of the paper. This should be clarified. The supplemental material extends the analysis to selected disciplines. Regarding these, the final section states "These results are limited to economics in the US and differences by field can be observed". Here it is unclear if this is supposed to mean that none of the results could be reproduced in the other disciplines or what the variations are.

Response: We thank the reviewer for highlighting these two important omissions. We added the clarification of the national scope (US) of this work in the methods section. We also added further analysis of the supplementary models for other disciplines in the results section:

“When we run an equivalent model on other fields (see Fig. S4) we observe both regularities and variations. Across all fields, a zero-distance in the collaboration network (self-citations) is a very important predictor of a citation link. In some fields such as Dentistry, Ophthalmology, Orthopedics, or Meteorology, further steps in the collaboration network are not as important, as we observe a discrete jump in the AME of step 0 with respect to steps 1 to 5. In other fields such as Mathematics, Probability, Management or Education, there is a smooth decrease in the AME of the different steps, forming a ladder of decreasing importance. The interpretation of these differences should be done with caution, as the discrete jumps observed in some fields could also be an indication of ill-defined collaboration networks, as size differences can be observed on the distribution of distances in the networks (see Fig. S2). Semantic similarity consistently plays a very significant role and ranks among the top AME of each model. Astronomy and Astrophysics in particular are an exception, as they have an ill-defined model for collaboration distances. This can likely be explained by the minimal overlap in the distributions of citing and non-citing document pairs within the collaboration network (see Fig. S2).”

And the discussion section:

“These results are limited to economics in the US, our analysis on other fields showed comparable results on semantic similarity (Fig. S3) with all fields, and more variability on social proximity (Fig. S2 & S4), given the complexity of building a comprehensive collaboration network on a single field.”

The authors have chosen one particular operationalization of social proximity, the distance between authors in the co-authorship network. They mention other possible operationalizations ("there are other ways of social proximity, such as working at the same institution, networking interactions on conferences or social events, or ties outside academia, that are not considered"). Their chosen operationalization certainly makes sense in terms of face validity and in feasibilty of data collection. However, it is very important to proffer convincing evidence that this indicator also covers the construct of social proximity sufficiently well to be useful for the study. This could be done by discussing the literature on academic social proximity, its ideal measurement operationalization, and the comparative performance of the co-authorship network measure, if such literature exists. The authors could additionally demonstrate the robustness of their method by showing that a minor variation in implementation does not alter the result patterns. The present implementation considers only the closest author pair connection but there are obvious alternatives such as the average or the sum of all observed pairs.

Response: By reviewing the literature that faced a similar type of operationalization problem, we have found that using the minimum distance between the groups of authors of the two documents is a standard practice. We added this clarification onto the methods section:

“Similar approaches that measure the social distance between two papers as the minimum distance between the groups of their authors have been used in previous literature (Martin et al., 2013; Wallace et al., 2012). Alternatively, we have also tested the use of the average distance between authors of the papers’ dyad and obtained consistent results.”

We also tested our methodology with the variation proposed by the reviewer, building a model with the average distance between the groups of authors instead of the minimum. It is worth noting that in order to make a comparable model (and therefore test the robustness of our approach) we needed to round and discretize the distance, which becomes a continuous variable when applying the mean instead of the minimum. We added the new results as Fig. S6 with the following interpretation in the results section:

“As an alternative operationalization, Fig. S6 shows the results for a model with the average distance (rounded and discretized to keep a similar model structure) between the authors of the two documents instead of the minimum.

The results remain robust despite this variation, although some expected differences emerge. Since step 0 corresponds to a self-citation for all authors, the proximity between the two documents is even greater, which also results in a higher AME value. In general, because in this alternative model each step indicates a closer proximity, all the AME values tend to be higher.”

Also, an argument could be made against the inclusion of self-citations into the variable for social proximity as there is nothing social about one's identity with oneself. That is to say, it is not logical that self-citations are one extreme of a continuous spectrum from close to distance social relationships.

Response: we agree that further justification was needed for the inclusion of self-citations. In our opinion, including self-citations is crucial, because our hypothesis is that the social proximity operates on the literature search because we are more familiar with the work of our closer colleagues. In this sense, self-citations are an extreme case, precisely because the work we are expected to be more familiar with is our own. We extended this rationale in the introduction:

“Including self-citations as an extreme case of social distance can be debated, since it involves a relationship with oneself. Our reasoning for including self-citations is that social proximity facilitates the literature retrieval process. We are typically more familiar with the work of colleagues we know well (Milard & Tanguy, 2018), and that includes our own. Therefore, self-citations are generally included in similar research studies (Martin et al., 2013; Singh et al., 2020), eventually leading to the presentation of collaboration networks as a broader extension of self-citation analysis (Wallace et al., 2012).”

For the prestige measure, the authors use the total citation count of the most cited author of a (cited) paper. In my opinion, there might be an issue of dependence between this predictor and the independent variable, the presence/absence of a citation link, because both variables are calculated from the same citation network. For a more generally highly cited author, it would appear that also any given paper would have higher chance of being cited by the given citing paper in a sampled paper pair. This would bias the analysis. I would like to ask the authors to address this possible concern.

Response: We thank the reviewer for this valuable comment. Indeed there was a dependency problem with this co-variable. We have addressed this by removing the cited paper from the sum of citations. This change indeed affected the outcome of the model, where now the AME of prestige lies between the AME of steps 4 and 5, instead of between steps 3 and 4. For the interactions (Figure 2b) the effect of prestige is also diminished and now only plays a marginal role when compared to social and semantic distances. We have updated the interpretation of results accordingly.

It might be interesting to compare and contrast the obtained results with findings of research that studied the influence of similar constructs on the outcomes of peer reviews.

Response: We thank the reviewer for this suggestion. After working on the extended literature review, we have not found any article to which we could compare our results in the context of peer reviews, with the exception of (Dondio et al., 2019), which we included in the introduction.

Reviewer #2: This study investigates how factors beyond intrinsic research quality—such as prestige, social proximity, and semantic similarity—influence citation patterns. Using a large dataset of disambiguated authors and citation links in U.S. economics, it finds that collaboration ties are the strongest predictor of citations, followed by semantic similarity. While prestige explains highly cited papers, most citations are driven by intellectual and social proximity. These findings highlight that citation inequalities stem more from network structures and research topics than from cumulative advantage, offering critical insights for science policy and evaluation.

This study integrates social proximity, semantic similarity, and prestige into a unified analytical framework. By combining large-scale disambiguated citation and collaboration networks with document-level semantic analysis, it offers a more comprehensive and empirically grounded understanding of citation mechanisms. The findings have important implications for the measurement and evaluation of scientific impact, particularly in policies related to hiring, research funding allocation, and academic promotion.

The paper could be further improved in the following aspects:

Response: We thank the reviewer for these comments, we address each of them below.

1. The paper needs to provide a precise definition of its core concepts. The term citing behaviour encompasses not only citation outcomes but also various aspects such as citation motivations. Therefore, it is necessary for the paper to clearly delineate the scope of this concept.

Response: We agree that the use of citing behaviour was vague, as we do not operationalize the motivations behind citations. We have therefore changed the title to "Citation proximus: the role of social and semantic ties on citations” to avoid confusion

2. To manage the computational cost of processing the full citation and author networks, the analysis is limited to the field of economics. However, this raises concerns about the representativeness of the findings—are the conclusions drawn from this dataset also applicable to other disciplines with low interdisciplinarity? Moreover, in highly interdisciplinary fields, do the relationships between citations and factors such as social proximity, semantic similarity, and prestige differ significantly from those in less interdisciplinary domains? The paper could benefit from including empirical analyses across more fields.

Response: In the supplementary information we have conducted a similar analysis over 12 fields that shared with Economics a high degree of isolation (operationalized as those that receive more than 50% of the citations from within the field). These results were not sufficiently developed in the manuscript in the original submission, and we have expanded the analysis on the results (see above in this letter). These results show different degrees of correspondence with those shown by economics, where the main differences seem to arise from the capability to properly represent distances in the collaboration networks. For more interdisciplinary fields this challenge would be even greater, as the collaboration network, as operationalized in the present manuscript, would be ill-defined. The challenge therefore in our opinion is to build a complete collaboration network of the full bibliometric database in order to properly define distances, beyond the characteristics of the field. This would allow for a more nuanced comparison between fields. Currently, the authors are working on this challenge, which implies a completely different approach to the problem. We have added this in the discussion section as future lines of work:

“One major challenge is to build a global collaboration network that includes all authors and articles across disciplines and countries to accurately compute distances between authors.”

3. The study uses multilingual-e5-large-instruct as a zero-shot classifier to generate 1024-dimensional embeddings for document representation. It remains unclear why other popular text embedding methods, such as BERT or SciBERT, were not considered. A comparison or justification for this choice would strengthen the methodological rigor.

Response: We agree that this needed more justification. This model was chosen because at the time of training it was the state of the art, according to huggingface leaderboard for equivalent tasks. We added the following clarification on the manuscript:

“This model was chosen instead of more traditional models such as BERT (Devlin et al., 2019) or models tailored for scientific documents such as sciBERT (Beltagy et al., 2019) because at the moment of training it was the best performing model at Huggingface leaderboard. Other possible models like SPECTRE (Cohan et al., 2020) are based on the ci

---

## [Decision Letter · Decision Letter 1]

10 Oct 2025

Citation proximus: the role of social and semantic ties on citations

PONE-D-25-35008R1

Dear Dr. Kozlowski,

We’re pleased to inform you that your manuscript has been judged scientifically suitable for publication and will be formally accepted for publication once it meets all outstanding technical requirements.

Kind regards,

Mu-Hsuan Huang

Academic Editor

PLOS ONE

Additional Editor Comments (optional):

Reviewers' comments:

Reviewer's Responses to Questions

**Comments to the Author**

Reviewer #1: All comments have been addressed

Reviewer #2: All comments have been addressed

2. Is the manuscript technically sound, and do the data support the conclusions?

Reviewer #1: Yes

Reviewer #2: Yes

3. Has the statistical analysis been performed appropriately and rigorously?

Reviewer #1: Yes

Reviewer #2: Yes

4. Have the authors made all data underlying the findings in their manuscript fully available?

Reviewer #1: No

Reviewer #2: Yes

5. Is the manuscript presented in an intelligible fashion and written in standard English?

Reviewer #1: Yes

Reviewer #2: Yes

Reviewer #1: The authors have thoroughly and comprehesively addressed the raised issues and concerns and further improved their study.

Reviewer #2: The authors have effectively addressed prior comments, improving clarity and depth.

The expanded related work and methodological clarifications on social proximity, semantic similarity, and prestige enhance rigor, while robustness checks strengthen validity. No ethical concerns were identified.

Overall, the manuscript is improved and suitable for publication.

**Do you want your identity to be public for this peer review?** For information about this choice, including consent withdrawal, please see our Privacy Policy

Reviewer #1: No

Reviewer #2: No

---

## [Editor Report · Acceptance letter]

PONE-D-25-35008R1

PLOS ONE

Dear Dr. Kozlowski,

I'm pleased to inform you that your manuscript has been deemed suitable for publication in PLOS ONE. Congratulations! Your manuscript is now being handed over to our production team.

Kind regards,

on behalf of

Professor Mu-Hsuan Huang

Academic Editor

PLOS ONE